# The coupling of the hydrated proton to its first solvation shell

**Markus Schröder** [1] ✉, **Fabien Gatti** [2], **David Lauvergnat** [3],
**Hans-Dieter Meyer**[1] **& Oriol Vendrell** [1] ✉

The Zundel ($H_5O_2^+$) and Eigen ($H_9O_4^+$) cations play an important role as intermediate structures for proton transfer processes in liquid water. In the gas phase they exhibit radically different infrared (IR) spectra. The question arises: is there a least common denominator structure that explains the IR spectra of both, the Zundel and Eigen cations, and hence of the solvated proton? Full dimensional quantum simulations of these protonated cations demonstrate that two dynamical water molecules and an excess proton constitute this fundamental subunit. Embedded in the static environment of the parent Eigen cation, this subunit reproduces the positions and broadenings of its main excess-proton bands. In isolation, its spectrum reverts to the well-known Zundel ion. Hence, the dynamics of this subunit polarized by an environment suffice to explain the spectral signatures and anharmonic couplings of the solvated proton in its first solvation shell.

The transfer of a hydrated proton between water molecules in an aqueous solution is accompanied by the large-scale structural reorganization of the environment as the proton relocates, giving rise to the Grotthus mechanism[1].

Due to the complexity of the liquid phase, the infrared (IR) spectroscopy of protonated water clusters in the gas phase opens a unique window to characterize and understand the elusive structural dynamics of these species. For example, the IR spectrum of the Zundel cation ($H_5O_2^+$) exhibits a prominent Fermi resonance in the ≈1000 cm⁻¹ spectral region of the shared-proton mode due to its strong anharmonic coupling with a combination of the wagging (water pyramidalization) and the oxygen–oxygen distance of the two flanking water molecules[2]. This important feature, key to understanding the strong coupling of the shared proton to its environment, could only be unambiguously measured following the development of accurate messenger spectroscopy (based on Neon tagging) of the gas-phase cation[3]. The theoretical assignment of this feature was a computational tour de force only possible due to the availability of a high-quality potential energy surface[4] in combination with full-dimensional (15-dimensional) quantum-dynamical calculations based on the multi-configuration time-dependent Hartree (MCTDH) approach[2,5–8].

Recent measurements of the IR spectrum of the Eigen cation ($H_9O_4^+$) reveal a strong coupling between the O–H stretch modes of the central hydronium unit with the water molecules in its first solvation shell. More importantly, they reveal strong shifts of the spectral position of the core O–H stretch modes caused by the polarization through the tagging agent in the second solvation shell[9]. The strong coupling with the first solvation shell leads to a large broadening of the core O–H stretch band, now spanning about 500 cm⁻¹ and markedly blue-shifted toward 2600 cm⁻¹ in comparison with the shared-proton band of the Zundel cation. The unambiguous characterization of this very broadband has remained a long-standing challenge[9] Yu and Bowman proved that the measured spectrum in ref. 9 can indeed be attributed to the Eigen isomer. Furthermore, they showed that the broad O–H stretch feature involves multiple states of the entire hydronium core[10] and that in addition the O–O stretching and O–H bending motions play an important role in the broadening of the O–H stretch band[10,11].

In particular, our analysis shows that the ligand waggings play an equally important role for the coupling of the excess proton to its solvation shell, both in symmetrically shared Zundel configurations[2] and in the Eigen-like form.

[1]Theoretische Chemie, Physikalisch-Chemisches Institut, Universität Heidelberg, Im Neuenheimer Feld 229, 69120 Heidelberg, Germany. [2]Université Paris-Saclay, CNRS, Institut des Sciences Moléculaires d'Orsay UMR 8214, 91405 Orsay, France. [3]Université Paris-Saclay, CNRS, Institut de Chimie Physique UMR 8000, 91405 Orsay, France. ✉e-mail: markus.schroeder@pci.uni-heidelberg.de; oriol.vendrell@uni-heidelberg.de

These findings were supported by detailed calculations of the linear absorption spectrum of the Eigen complex with different levels of theory, most successfully using a combination of QCMD and VSCF/VCI methods[10–14]. In this paper, we simulate the linear absorption spectrum of the Eigen cation for the first time using full-dimensional (33D) quantum-dynamical calculations using polyspherical coordinates that are adapted to the Eigen motif. These allow for the accurate inclusion of correlations between low frequency, large amplitude displacements, and the O–H stretch and other higher frequency modes. Our spectra, based on the Yu–Bowman PES first published in ref. 12 that was also applied to clusters with up to 21 water molecules[15], are in excellent agreement with the available messenger-tagging spectra in the full spectral range between 0 and 4000 cm$^{-1}$ (see ref. 9). We compare the full spectrum of the Eigen cation with those calculated with frozen subsets of degrees of freedom all the way down to a dynamical (polarized) $H_5O_2^+$ subunit embedded in the static scaffold of the remaining Eigen cation. This analysis reveals that the underlying coupling mechanism of the solvated proton with its first solvation shell is strikingly similar in both the Zundel and Eigen forms: a dynamical subunit formed by two water molecules and a proton is the least common denominator structure that reproduces the spectrum and anharmonic mode couplings of the Zundel and Eigen forms depending on the conformation of its static environment. Along this analysis, we confirm existing assignments[9–13,16] of various peaks in $H_9O_4^+$. We would like to stress that theoretical absorption-band assignments have already been reported in ref. 10 and are not the main focus of this contribution. We contribute two assignments for hitherto unknown features in the low-frequency region, where no experimental data is currently available.

## Results

### IR spectrum of the Eigen cation

Figure 1 shows the calculated absorption spectrum of the Eigen cation $H_9O_4^+$ in comparison with the experimental spectra from refs. 13 and 9. The calculated IR spectra are based on a 33D quantum-mechanical description of the Eigen cation. Such simulations could be achieved only after the unique combination of recent developments in our groups; They constitute the largest quantum wavepacket simulations of a flexible molecular system using a general potential energy surface and curvilinear coordinates reported to date. Details of the 33-dimensional quantum-dynamical calculations, including the construction of the kinetic[17] and potential[18] energy operators, and the wavefunction propagations with the multilayer MCTDH method[19–21], are provided as supporting information.

The calculated spectrum is red-shifted by 70 cm$^{-1}$ to match the main features of the experimental spectrum. The shift originates from the fact that we obtain the ground-state energy and the spectrum from separate calculations. The ground-state wavefunction has a much simpler structure than the time-evolved one and it is hence better converged. This explains the global shift. The spectrum is obtained as the average over the spectra corresponding to the three polarization directions of light with respect to the molecular frame, thus considering the random orientation of the molecules in the experiment (see "Methods" and extended data for details).

The overall agreement of calculated and experimental spectra is very good although the resolution of the calculated spectrum is ~30 cm$^{-1}$ and limited by the 1 picosecond duration of the dipole–dipole correlation function. The calculated peak positions are listed in Supplementary Table 1 of the supporting material alongside with experimental results and assignments. In particular, the substructure of the broad core O–H stretch band and practically all features of the spectrum are reproduced in comparison with the tagging-agent IR measurement. Our simulations thus further support the interpretation that (i) the spectra in refs. 9 and 13 correspond to the triply-coordinated hydronium form of $H_9O_4^+$ stoichiometry, and (ii)

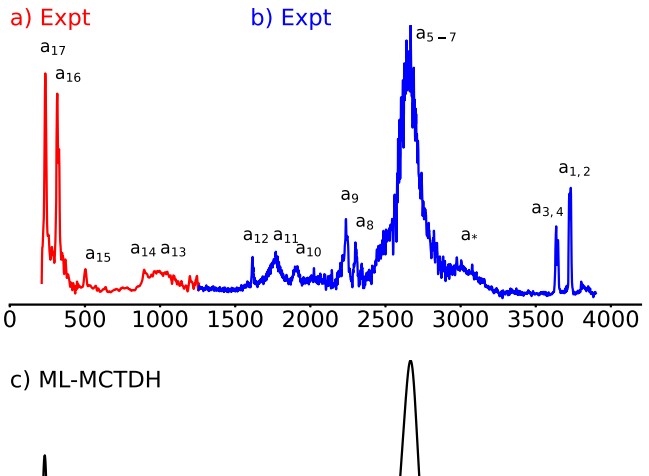

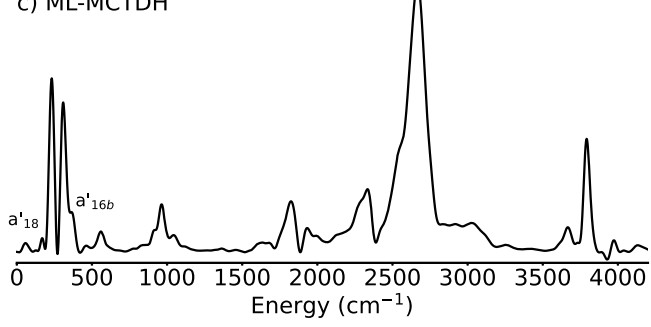

**Fig. 1 | Absorption spectrum of the Eigen Cation $H_9O_4^+$. a** Experimental spectrum from ref. 13. **b** Experimental spectrum from ref. 9. **c** Calculated spectrum (red-shifted 70 cm$^{-1}$ to match experimental line positions). Both experimental spectra detected via photodissociation of $D_2$ tagged clusters. The assignments of the peaks follow the nomenclature of refs. 9, 13 and are discussed in Supplementary Table 1. Source data are provided under https://doi.org/10.5281/zenodo.7064870.

that the $D_2$ tagging agent negligibly alters the spectrum of $H_9O_4^+ \cdot D_2$[9,13] compared to $H_9O_4^+$.

### Deconstructing the broad hydronium O–H stretch band

The key to understanding the anharmonic couplings of the core O–H stretch modes to their first solvation shell lies in characterizing the broadening and composition of the main core O–H stretch band in pristine $H_9O_4^+$: This feature carries most of the IR intensity related to the coupled motions of the central proton stretching modes.

While studying the broad O–H stretch peak, Duong et al.[11] found that this band is characterized by many highly entangled eigenstates in terms of normal-mode excitations. In the theoretical part of their work, Duong et al. used VSCF/VCI calculations involving the hydronium core modes, O–O stretch and O–H bending modes to identify states contributing to the broadening. Here, we take a different approach and deconstruct the formation of this band by first freezing all modes of the Eigen cation, except those of the hydronium core, to their expectation values, and then by successively bringing back the environment. The spectra obtained in this way are shown in Fig. 2. They correspond to the z-component of the dipole moment (the polarization is aligned with one of the hydronium hydrogen bonds), since this is the component responsible for the largest response of the core O–H stretch modes. Freezing specific coordinates is achieved by removing all differential operators of a frozen coordinate from the Hamiltonian in a Hermitian way and by fixing their position to the corresponding expectation value in the ground vibrational state of the full-dimensional system.

The IR spectrum of the hydronium core embedded in the frozen environment (cf. Fig. 2a) has a very simple structure. The vibrational eigenstates corresponding to the two sharp peaks near 2700 cm$^{-1}$ were obtained by full diagonalization explicitly: The dominant peak corresponds to the hydronium core O–H stretch mode, whereas the smaller structure corresponds to an out-of-plane excitation of the central

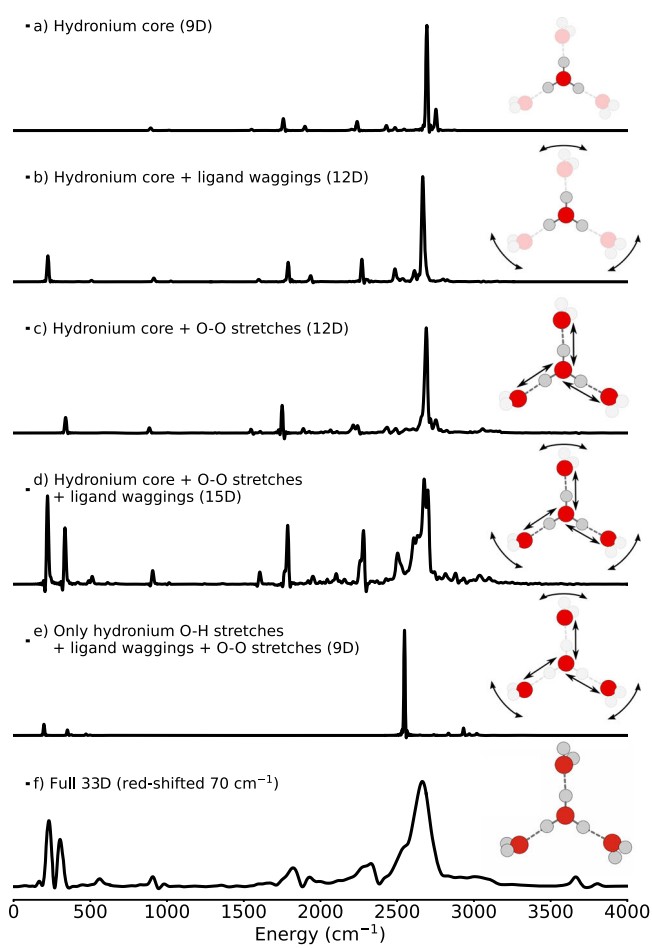

Fig. 2 | **Spectra obtained with the z-component of the dipole moment (cf. Eq. (2)) for various reduced models.** The dimensionality (9D, 12D etc.) denotes the number of active coordinates. Other coordinates are frozen to their expectation value positions for the vibrational ground state. Correlation time in **a**–**e**: 2000 fs, (**f**): 1000 fs. Source data are provided under https://doi.org/10.5281/zenodo.7064870.

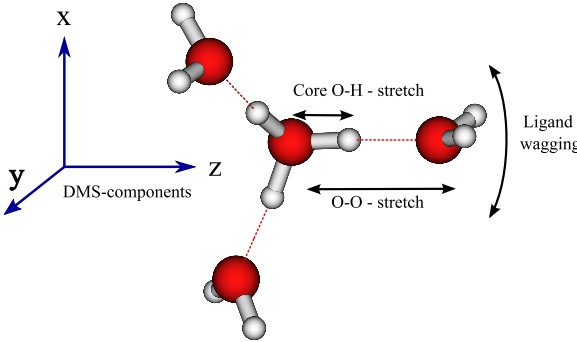

Fig. 3 | **Illustration of ligand wagging, core O–H stretch and O–O stretching motion of $H_9O_4^+$ exemplary in one of the three arms of the cation.** Note that in the ligand-wagging motion only the two hydrogen of the outer water molecules move as indicated by the arrow. The coordinate system on the left indicates the directions of components of the dipole moment surfaces (DMS).

gains significant intensity as well. This structure coincides with the spectral position of the low-energy shoulder of the broadband in the full spectrum. Moreover, now the low-intensity background on the high-energy shoulder at around 3000 cm$^{-1}$ emerges. VSCF/VCI analysis[11,13] attributed this to a combination mode of hydronium O–H stretch and O–O strecthing modes. This assignment is fortified in Fig. 2c, where a peak at 3000 cm$^{-1}$ appears while only the hydronium core and the O–O stretches are modeled. Adding the ligand-wagging modes then leads to the diffuse signal observed in the experimental spectrum.

In the spectra in Fig. 2, panels a to d, the hydronium core retains its full mobility. The question arises, whether only proton displacements parallel to the hydrogen bonds are important, or whether displacements perpendicular to the hydrogen bonds also contribute to the main proton-transfer band. These perpendicular displacements span the hydronium bending, wagging, and pyramidalization modes. Freezing the perpendicular displacements of the hydronium protons (panel e) has dramatic consequences. The spectrum is now dominated solely by the proton-transfer peak. Peaks of the ligand wagging and O–O stretching fundamentals are again visible with low intensity at low energies, as well as peaks at -3000 cm$^{-1}$ that are combinations of hydronium core O–H stretch, ligand wagging, and O–O stretch modes. However, the inability of the three central protons to move perpendicular to the hydrogen-bond directions has largely suppressed their coupling with the first shell of ligand water molecules. Crucially, no broadening of the O–H stretch peak is present, as opposed to the spectrum in panel d. This leads to the conclusion that the vibrational eigenstates spanning the broad hydronium core O–H stretch band correspond to combinations and overtones of the central O–H stretch modes with O–O stretch displacements, hydronium bending and hydronium wagging, and ligand waggings, whereby none of those coupled hydronium and environment modes can be removed. A full characterization of the vibrational eigenstates in terms of quantum numbers of some basis of uncoupled vibrational modes is currently out of reach due to the very high density of vibrational states in the spectral region of the band and the high dimensionality of the problem.

## The dynamical $H_5O_2^+$ subsystem

We have deconstructed the main hydronium O–H stretch band. It originates from the anharmonic couplings of the center O–H stretch modes with perpendicular modes of the central hydronium and modes involving the O–O stretchings and waggings of the three surrounding water ligands. The question now arises: are the three water molecules in the first solvation shell of the Eigen cation necessarily involved in explaining the coupling mechanism, spectral position, and width of

hydrogen atoms. The peaks near 1800 and 2300 cm$^{-1}$ correspond to other modes of the hydronium core also seen in the full spectrum and agree with the assignments in refs. 9, 10, 12, 13.

Adding either wagging modes of the outer water molecules (Fig. 2b), or O–O distances, (Fig. 2c), leads to the appearance of their fundamental modes in the spectrum (Illustrated in Fig. 3). In the latter case, some peaks on the low-energy shoulder of the main O–H stretch peak gain some intensity. Moreover, with the inclusion of the O–O stretching coordinates, two small peaks appear at 2300 cm$^{-1}$ correlating with $a_8$ and $a_9$ in the full-dimensional spectrum. Apart from this, the overall structure of the spectrum changes only slightly. In particular, there is no significant broadening of the O–H stretch peak.

More complex spectral features emerge when adding both solvation-shell water wagging modes and O–O distances together (Fig. 2d). Now, the spectrum is not the simple sum of the previous two panels and cannot be explained by the fundamental modes of the involved coordinates alone. The broad hydronium core O–H stretch band centered at 2700 cm$^{-1}$ is now composed of at least four separate contributions with significant intensity. (Here, we note that all spectra are normalized to unity maximum height such that with the O–H stretch peak now decomposing into multiple smaller peaks the relative height of all other peaks increases.) Two of those peaks, contributing to the low-energy shoulder of the central peak at ~2600 cm$^{-1}$, have gained significant intensity. Finally, a peak slightly above 2500 cm$^{-1}$

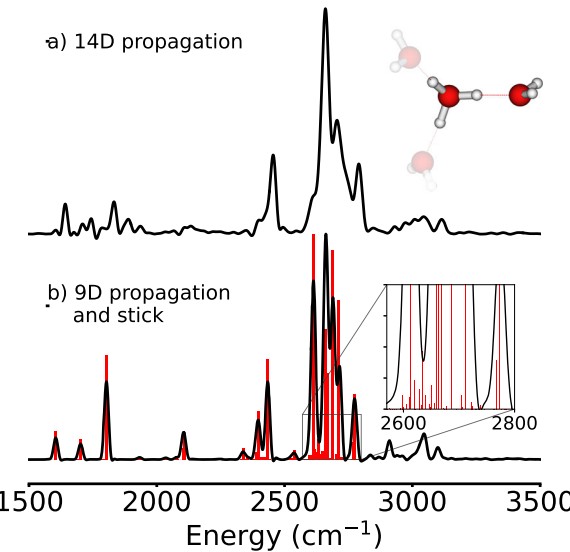

**Fig. 4 | Spectra for reduced H₅O₂⁺ models.** Spectra obtained with the z-component of the dipole moment surfaces for reduced $H_5O_2^+$ models by freezing modes to positions corresponding to their expectation values for $H_9O_4^+$ **a** obtained with a dipole–dipole-correlation function of 2000 fs using a 14D model, **b** obtained with a dipole–dipole-correlation function of 2000 fs using a 9D model (black curve), and obtained as a stick spectrum using eigenstates (red lines). Source data are provided under https://doi.org/10.5281/zenodo.7064870.

the main proton-transfer band? Alternatively, can a smaller dynamical subunit completely account for the properties of the first solvation shell of the solvated proton? The hydronium cation ($H_3O^+$) can be discarded as the least common denominator subunit by comparing Fig. 2a and f. Even though the O–H stretch peak in Fig. 2a is in the correct position, the broadening is not observed.

Instead, we consider one $H_5O_2^+$ subunit, that can be understood as a polarized Zundel cation (14 coordinates) and freeze all internal, angular and relative coordinates of the two other water ligands to their corresponding expectation values positions (cf. Fig. 4), as well as the two free-standing hydronium O–H stretches, as those do not interact with their immediate environment dynamically any more. For comparison, we also consider a reduced version of the $H_5O_2^+$@Eigen cation where the rocking, relative water rotation and internal modes of the external water are also frozen, thus yielding a nine-dimensional system for which the lowest 250 eigenstates can be computed with the improved relaxation algorithm (ticks in Fig. 4b)[22,23].

The IR spectrum of the dynamical $H_5O_2^+$@Eigen cation is strikingly similar to the full Eigen cation spectrum, as seen in Fig. 1. The main O–H stretch band presents a comparable broadening and is centered at the same frequency. Other flanking peaks appear at the correct positions as well. The analysis of 1D and 2D probability densities of the calculated eigenstates of the nine-dimensional model reveal that the vibrational states that participate in this band are complex combinations and overtones of the same vibrational coordinates previously found to contribute to the broadening of the core O–H stretch band in the Eigen cation. Just pulling the external water molecules by about 0.5 Å away from the central hydronium, while leaving them frozen, results in a shift of the O–H stretch band to the red by about 600 cm⁻¹ (cf. supporting material) as well as a reduction of the ground-state expectation value of the O–O distance by 0.1 Å and an increase of the O–H distance expectation value by 0.06 Å. This indicates the extreme sensitivity of the position of this band to the polarization by the first solvation shell of water molecules. This trend has also been observed in similar studies on protonated water clusters.[24–28] Pulling the waters further to infinity leaves the bare

Zundel cation with its O–H stretch band red-shifted by about 1600 cm⁻¹ compared to the Eigen cation[2].

Based on these observations, we argue that two protonated water molecules, nominally the polarized $H_5O_2^+$/Zundel subunit, constitute the dynamical least common denominator structure explaining the anharmonic couplings and spectral signatures of the solvated proton in its first solvation shell. This statement does not concern the relative population of the Zundel and Eigen structures in solution, which has been investigated separately by Marx et al. using path integral techniques, cf. ref. 1.

In isolation, the main shared-proton peaks in the Zundel cation are strongly red-shifted compared to the Eigen cation. The shared-proton motion strongly couples to the wagging (pyramidalization) of the two water molecules and to the O–O stretching mode, and results in the well-characterized Fermi resonance doublet centered at about 1000 cm⁻¹ [5,6,29,30]. Embedded in the potential of two flanking, frozen water molecules, the polarized $H_5O_2^+$@Eigen subsystem features its O–H stretch band at the same spectral position as the full-dimensional Eigen cation, i.e., blue-shifted to about 2600 cm⁻¹ because the shared proton is now much closer to the central water molecule. The broadening of the core O–H stretch band in the polarized $H_5O_2^+$@Eigen and Eigen cations is strikingly similar. Our simulations demonstrate that the same set of vibrational coordinates and corresponding combined excitations are responsible for the strong coupling of the shared proton to the rest of the scaffold in the Zundel[5,6,29,30], polarized $H_5O_2^+$@Eigen, and Eigen cations. These effects are strongly cooperative as opposed to additive. These relevant coordinates are the hydronium O–H bending and wagging modes, the ligand water wagging modes, and the O–O hydrogen-bond stretching mode.

## Discussion

This work has provided a set of full-dimensional quantum simulations of the Eigen cation based on flexible, curvilinear coordinates and a very accurate potential energy representation. The simulated IR spectra cover the chemically relevant spectral range between 0 and 4000 cm⁻¹ with one single time-propagation of a highly correlated multi-configurational wavefunction. The spectra extend below the smallest frequency accessible experimentally using ion tagging techniques and reveal the signatures of very low-frequency, global vibrational modes. Both the Zundel and Eigen cations feature very prominent spectral features related to the anharmonic couplings of the hydrated proton with its first solvation shell. In the Zundel cation, a strongly red-shifted double peak[6,30] originates from the fundamental vibration of the equally shared proton at about 1000 cm⁻¹. This doublet is a Fermi resonance that involves the wagging modes of the flanking water molecules as well as the hydrogen-bond O–O stretching. The isolated Eigen cation, instead, features a very broad band at 2600 cm⁻¹ with little resemblance to the shape and position of the Zundel's double peak. Nonetheless, a careful analysis reveals that similar anharmonic couplings compared to the Zundel form are involved in the broad Eigen cation band, namely the hydronium and ligand waggings to the largest extent, combined with hydrogen-bond stretchings. Indeed, the hydronium waggings are crucial to the coupling mechanism: freezing the central hydronium waggings in a flexible first solvation shell results in a simpler IR spectrum than when considering a fully flexible hydronium in a frozen environment (cf. Fig. 2a, e).

Based on these results and observations, we arrive at a key insight: two dynamical water molecules and a proton, i.e., a $H_5O_2^+$ subunit embedded in the remaining frozen scaffold of the Eigen cation, present all anharmonic couplings and spectral signatures of the fully flexible Eigen cation in the region of the main proton-transfer band. Depending on its environment, the $H_5O_2^+$ subunit can describe both the spectrum of the Zundel and Eigen cations. For this effect, it is sufficient that two frozen, hydrogen-bond acceptor water molecules polarize the dynamical $H_5O_2^+$ subunit that constitutes the proton's first

solvation shell. This finding, backed by our full quantum-mechanical approach, is suggestive of picturing the Eigen cation as three over-lapping and strongly polarized $H_5O_2^+$ subunits in the spirit of the classical "special pair dance" models of the solvated proton[31–33].

The question of whether the proton forms an Eigen or Zundel cation in aqueous acid solutions has given rise to many studies even recently: some new experimental works[34–36] have suggested that the population of the Zundel cation is larger than previously thought. On the other hand, new simulations[33] have interpreted these experimental findings in an opposing sense pointing to a dynamic Eigen cation as the most prevalent hydrated proton species. Our new results do not bring information about the relative populations of the two structures but stress that the difficulty to solve the problem may partly come from the fact that the $H_5O_2^+$ subunit can exhibit very similar spectral sig-natures compared to the Eigen cation when placed in a polarizing environment. Establishing these structural and dynamical relations on the basis of full-dimensional quantum dynamics is an important direction for future work. The computational and theoretical devel-opments reported in this work may be decisive when approaching even larger and more complex systems.

## Methods

### High-dimensional quantum dynamics

The full-dimensional (33 vibrational degrees of freedom) quantum-dynamical description of the IR spectrum of the Eigen cation requires the combination of various technologies that have been developed and integrated into the software packages maintained in our research groups. These technologies relate to the three main obstacles that stand on the way towards a full quantum-dynamical description of anharmonically coupled, flexible, and high-dimensional vibrational problems.

(i) Describing flexible and anharmonic systems, e.g., with several equivalent minima in their potential energy surface (PES), requires the use of chemically meaningful coordinates such as bond lengths, bond angles, and dihedral angles. The use of adequate coordinates enormously facilitates the numerical representation and con-vergence of the vibrational wavefunctions in high dimensions. The price to pay, though, is the very lengthy and complicated expression for the corresponding kinetic energy operator (KEO). For the Eigen cation, the exact, analytic KEO has a total of 4370 terms, and its manual derivation becomes de facto intractable. Some of us and others have therefore developed a completely systematic method to set up the KEO for a specific family of internal molecular coordinates: the polyspherical coordinates[37–39]. This method is implemented in the TANA software, which provides analytic expressions of the kinetic energy operator in a machine-readable format[17,40,41]. Very importantly, TANA also provides numerical library routines to per-form forward and backward transformations between the Cartesian coordinates of the atoms and the internal coordinates of the mole-cule, which are needed when setting up the potential energy operator in these internal coordinates.

(ii) The second obstacle is the so-called "curse of dimensionality" for representing and storing the wavefunction of the system: the number of possible quantum states of the system (e.g., given as the amplitudes on quadrature points in coordinate representation) grows exponentially with the number of physical coordinates. Without an efficient data reduction scheme one would be limited to model up to about six internal degrees of freedom of a molecule, corresponding to about four atoms (neglecting rotations). To overcome the curse of dimensionality, the state vector needs to be stored and processed in a very compact form. To this end, we employ the multilayer multi-configuration time-dependent Hartree algorithm,[19–21,42,43] which repre-sents the wavefunction as a hierarchical Tucker tensor-tree[44–46].

(iii) The solution of the time-dependent Schrödinger equation within this tensor format requires that also the system Hamiltonian is

expressed in a matching form. This can be, e.g., a sum of products of low-dimensional operators. The KEO in polyspherical coordinates always consists of sums of products of elementary functions and derivatives of single coordinates[39] (this is one of the main advantages of the polyspherical coordinates) and needs not be discussed further here. A more challenging task is to express the PES and, if needed, other surface-like operators such as dipole moment surfaces (DMS), in a matching format. The PES and DMS are usually made available as separate software libraries, and are often defined in the Cartesian coordinates of the atoms[10,12,16]. Most applications in our groups have relied until recently on the transformation of the PES into a Tucker format with the so-called Potfit algorithm[47–49], and its hierarchical multilayer variant[50]. This algorithm suffers from the curse of dimen-sionality because ultimately it requires a full representation of the primitive product grid in configuration space. Modifications of Potfit have been developed over the years to partially overcome this difficulty[51–53], making it possible to work with about 9–15 coordinates. This is clearly insufficient to approach a system of the size of the Eigen cation. A more recent development in surface re-fitting uses the so-called canonical tensor decomposition[54] (CP), also called PARAFAC or CANDECOMP in the literature[55,56]. Within the canonical format, orthogonality restrictions on the basis functions are relaxed such that a much more compact tensor representation can be achieved, at the cost, however, that the fit is much harder to obtain. This is usually achieved using an alternating least squares (ALS) algorithm that iteratively improves an initial guess tensor. The ALS algorithm in the original form requires to perform high-dimensional integrals as well. In a recent publication[18] Monte-Carlo integrations are used to perform the integrals. This not only mitigates the curse of dimensionality but also allows for importance sampling such that low-energy regions of the potential (where the wavefunction resides) can be fitted with ele-vated accuracy. This development has opened the path to obtain global but compact surface fits in a tensor format of high-dimensional potentials.

In essence, we developed and combined three technologies to be able tackle such a high-dimensional problem as the 33-dimensional Eigen cation: (1) the TANA software to obtain the KEO and to provide the coordinate transformations for the PES fitting; (2) PES fitting into a canonical tensor format using a Monte-Carlo version of the ALS algo-rithm; and (3) the multilayer MCTDH algorithm to solve the time-dependent Schrödinger equation. In the present contribution, we have used the highly accurate, full-dimensional PES, and DMS provided by Yu and Bowman[10,12,16]. The surfaces were re-fitted into a canonical tensor format using 2048 terms for the PES and 1024 terms for each of the three components of the DMS, respectively.

### Calculation of IR spectra

The linear absorption spectra that are compared to the experimental spectra are computed as averages of the spectra resulting from the three dipole moment components for the x-, y-, and z-directions as

$$I(\omega) = \frac{1}{3}(I_x(\omega) + I_y(\omega) + I_z(\omega)). \qquad (1)$$

The averaging mimics the random orientational distribution of the molecule in the experiment. The single components also shown in some figures below are calculated as[6]

$$I_j(\omega) \propto \omega \, \mathrm{Re} \int_0^\infty dt \left\langle \Psi_{\mu_j} | \Psi_{\mu_j}(t) \right\rangle \exp(i(\omega + E_0/\hbar)t), \quad j = x, y, z \qquad (2)$$

where $E_0$ is the ground-state energy and

$$\left| \Psi_{\mu_j} \right\rangle = \mu_j |\Psi_0\rangle \quad j = x, y, z \qquad (3)$$

is the vibrational ground state $|\Psi_0\rangle$ operated with $\mu_j$, one component of the dipole operator. The time-dependent state $|\Psi_{\mu_j}(t)\rangle$ is obtained by solving the time-dependent Schrödinger equation with initial value $|\Psi_{\mu_j}\rangle$.

## Assignments

To assign modes to the peaks a number of test states that contain zero-order excitations in selected modes have been created and cross-correlated with the dipole-operated and propagated ground state.

The Fourier transform of the resulting cross-correlation shows peaks only at frequencies where both, the test states and dipole-operated ground states populate the same eigenstate. The cross-correlation functions are defined as

$$C_{i,X}(t) = \left\langle \Psi_X | \Psi_{\mu_i}(t) \right\rangle \quad i = x, y, z \tag{4}$$

and $|\Psi_X\rangle = X|\Psi_0\rangle$ being the $X$-operated ground state with an operator $X$ as detailed below. The Fourier transformed of the cross-correlation is given as

$$F_{i,X}(\omega) = \, \propto \text{Re} \int_0^\infty dt \left\langle \Psi_X | \Psi_{\mu_i}(t) \right\rangle e^{i(\omega + E_0/\hbar)t} \quad i = x, y, z, \tag{5}$$

with $E_0$ being the ground-state energy. Note that, other than for the absorption spectra, no frequency prefactor $w$ is multiplied to the spectrum.

The test states $|\Psi_X\rangle = X|\Psi_0\rangle$ have been created by constructing the operator $X$ as linear combinations of position operators of specific coordinates. This creates a linear combination of wavefunctions, with nodes in the respective modes, hence resembling zero-order excitations which mimic the action of the dipole moment surface but restrict the action only to the aforementioned modes. We use the notation $q(+++)$, $q(-++)$ and $q(0-+)$ for $X$ in the test states. Here, the $q$ indicate physical coordinates and the string of signs in brackets identifies one of the three orthogonal linear combinations of the coordinates $q$ in the three "arms" A, B, and C of the Eigen cation (cf. Supplementary Fig. S8, extended data), where specifically

$$q(+++) := q_A + q_B + q_C \tag{6}$$

$$q(-++) := -2q_A + q_B + q_C \tag{7}$$

$$q(0-+) := -q_B + q_C \tag{8}$$

(with the exception of label $q = '\theta' = (q_A = \theta, q_B = \varphi_{AB}, q_C = \varphi_{BC})$, and $q = 'b'$ describing the ligand O–H bending as a linear combination of two Jacobi coordinates $b_{\{A,B,C\}} = -0.4\, r_{1,\{A,B,C\}} + 0.3\, r_{2,\{A,B,C\}}$. Similarly, the symmetric O–H stretching of the ligands is described by $q = 'v^{(s)}'$ with $v^{(s)}_{\{A,B,C\}} = 0.3\, r_{1,\{A,B,C\}} + 0.4\, r_{2,\{A,B,C\}}$, while the asymmetric O–H stretching $v^{(a)}_{\{A,B,C\}} = v_{\{A,B,C\}}$ is described by the Jacobi angle (cf. Supplementary Table S1 of assignments and Supplementary Table S2 and Supplementary Fig. S9 of coordinate definitions in the extended data section).

Non-vanishing cross-correlations hence show the existence of non-vanishing overlap of the dipole-operated state $\Psi_{\mu_i}$ and the test state characterized by a linear combination of single mode excitations of character Eq. (7).

## Kinetic energy operator

As for the Zundel cation[29], we adopted a mixture of Jacobi, Cartesian, and valence vectors. For each external molecule of water (in blue in Supplementary Fig. S7, extended data), we use two Jacobi vectors: one from one hydrogen atom to the other and one from the middle of $H_2$ to the oxygen atom. The central oxygen atom is linked

to the other oxygen atoms by three O–O valence vectors. The global $z$ Body-Fixed (BF) axis is parallel to $R_1^{BF}$, one of the O–O vectors. The groups $S^1$ and $S^2$ are gathered into two subsystems so that they have their own BF frame with the $z$ axis parallel to $R_2^{BF}$ or $R_3^{BF}$. The molecule at the top of Supplementary Fig. S7 (extended data) is also gathered in one subsystem with the $z$ axis parallel to the H–H vector. The same is true for the other two molecules of water, except that they define "subsubsystems" in $S^1$ and $S^2$. The three OH valence coordinates starting from the central oxygen atom are re-expressed in terms of Cartesian (and not spherical) coordinates to avoid singularities in the kinetic energy operator (KEO).

All the other vectors are parametrized by spherical coordinates in their BF frame. The rotation of each BF frame is parametrized by Euler angles. We follow the conventions of the general formulation for polyspherical coordinates[57] that is implemented in the TANA software[17,40]. The correctness of the implementation has been checked on many systems by comparing the KEOs with those obtained numerically with the TNUM software[41,58]. We thus obtain an exact operator. TANA provides the operator in an ASCII file that can be directly read by MCTDH. One advantage of the family of polyspherical coordinates is that it always leads to an operator in a sum of products of one-dimensional operators. In the present case, with those coordinates and their corresponding ranges, we avoid all the possible singularities in the KEO so that we do not need to use 2D DVRs that are numerically less efficient than products of 1D DVRs.

## Sum-of-products of potential and dipole moment surfaces

In the present case, the potential energy and dipole surfaces were made available to us in the form of a numerical library in ref. 12. The potential and dipole routines take a single coordinate vector as input and return the respective energy value or three-component dipole vector.

The Heidelberg MCTDH implementation[42,47,59–62] relies on an explicit numerical representation of the potential in terms of a sum of products of one- or low-dimensional functions that are sampled on a primitive grid. Hence, given a numerical library routine for the potential (and dipoles), a preprocessing step is necessary that creates the required numerical representation of the potential from the output of the library routines.

In the present case, the potential energy surface has been decomposed into a sum-of-products of 2048 low-dimensional terms, more precisely into a Canonical Polyadic Decomposition form. The low-dimensional basis functions are defined on the coordinates that correspond to those of the bottom layer of the wavefunction tree, (cf. Supplementary Fig. S10, extended data). Such a decomposition can be used within the Heidelberg MCTDH package. The decomposition was created using a Monte-Carlo variant[18] of the alternating least squares algorithm that is often employed to obtain canonical decompositions. In total, eight symmetries have been incorporated into the PES fit, all of them with respect to rotations of the outer water ligands. Other symmetries could not be implemented due to the mixing of coordinates. For details about the algorithm, the reader is referred to ref. 18.

The surface fit needs to be performed in the internal dynamical coordinates, the library routines usually require Cartesian coordinates to calculate the respective potential energy such that here we interlinked the TANA program with the fitting program to be able to transform between the two sets of coordinates.

## Data availability

The raw data for Figs. 1, 2, and 4 as well as all necessary input files and instructions compatible with the Heidelberg MCTDH package are provided to reproduce the infrared spectrum of the Eigen cation. These data are accessible under the URL https://doi.org/10.5281/zenodo.7064870[63].

## Code availability

The TANA and MCTDH codes with their full documentation and any further input files needed to reproduce particular results of the current contribution are available upon request from the authors.

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

## Acknowledgements

The authors thank Prof. Mark Johnson and Prof. Knut Asmis for sharing experimental data with us and Prof. Joel Bowman for the source code of the potential energy surface. We furthermore thank the High-Performance Computing Center in Stuttgart (HLRS) under the grant number HDQM_MCT as well as the bwHPC project of the state of Baden-Württemberg under grant number bw18K011 for providing computational resources. The authors thank the CNRS International Research Network (IRN) "MCTDH" for financial support. For the publication fee we acknowledge financial support by Deutsche Forschungsgemeinschaft within the funding programme "Open Access Publikationskosten" as well as by Heidelberg University. OV acknowledges funding by the Deutsche Forschungsgemeinschaft (DFG, German Research Foundation) – Project number 442507947.

## Author contributions

M.S., H.-D.M., and O.V. conceived the idea and planned the calculations and the analysis methodology. M.S. contributed to the SOP fitting of the PES and DMS and performed the dynamical calculations and analysis. F.G., D.L., and O.V. designed the coordinates system. F.G. and D.L. generated the corresponding analytical KEO. The text was initially composed by M.S. and O.V., and all authors contributed to the discussion and interpretation of the results and to the final version of the manuscript.

## Funding

## Competing interests

The authors declare no competing interests.
