## [Peer Review File · Nature Communications]

REVIEWER COMMENTS

Reviewer #1 (Remarks to the Author):

There is no question that these are state of art MCTDH calculations of the IR spectrum of H₉O₄⁺ using state-of-the art potential and dipole moment surfaces of Yu and Bowman and co-workers. However, the paper is weak in several key ways and these weaknesses need to be addressed.

First, the calculations are not put into proper scientific context in order to assess the scientific impact of the calculations. Indeed, there was controversy surrounding the interpretation (Zundel vs Eigen) of the experimental spectra of Johnson and co-workers of the H₉O₄⁺ cluster. The authors are apparently unaware that this controversy was settled in their ref. 13 "High-Level Quantum Calculations of the IR Spectra of the Eigen, Zundel, and Ring Isomers of H+(H₂O)₄" Find a Single Match to Experiment J. Am. Chem. Soc. 2017, 139, 10984–10987. Those calculations were carefully done using VSCF/VCI considering the Eigen, Zundel and Ring isomers and definitely found a match the Johnson experiment for the Eigen isomer. (See file attached) I guess this is conclusion of this paper. Well that's fine but it's not new. This is nice corroboration of that work. Figure 1 comparing the present calculations with experiment is impressive. However, a very similar comparison was already published using the same PES and DMS earlier in their ref. 13 and strangely this not noted. I reproduce a figure from the JACS paper below. Also there was no shifting of that VSCF/VCI spectrum, whereas we read in Fig. 1 caption "Calculated spectrum (red-shifted 70 cm⁻¹ to match experimental line positions). The authors need to discuss the cause of this 70 cm⁻¹ "offset" with experiment

This important controversy should be highlighted in the paper; however, the fact that it has now been definitively resolved reduces the scientific impact of this manuscript. The authors go on to write "In particular, we identify the anharmonic vibrational modes that ex26

plain the large broadening of the proton transfer peak in the experimental IR spectrum

27 of the Eigen cation, of which the origin remained so far unclear.3–5". Well this is not accurate. See below

Also, there is no evidence of a proton transfer occurring in the Eigen cation H₉O₄⁺, quite the opposite. See below.

Some strange wording,

1. "The proton transfer band of the Eigen cation was studied by Yu and Bowman⁴, 13 using a variational configuration interaction approach with up to a 4-mode representation, using normal modes, of the same PES we are using...". Well that PES and DMS is the earlier Yu-Bowman PES and

DMS that the authors are using. I suggest referring to the PES as the “Yu-Bowman PES” which was actually first reported in Yu, Q.; Bowman, J. M. Communication: VSCF/VCI Vibrational Spectroscopy of H₇O₃⁺ and H₉O₄⁺ using High-level, Many body Potential Energy Surface and Dipole Moment Surfaces. *J. Chem. Phys.* 2017, 146, 121102. This should be cited.

2. Continuing “ found that this band is characterized by highly entangled eigenstates in terms of normal-mode excitations, and that therefore these eigenstates cannot be assigned each to a well-defined vibrational configuration.”^{4, 13} This is not correct. That band was of course focused on in the papers using the Yu-Bowman PES and DMS in VSCF/VCI calculations. In particular, the authors need to cite and discuss “Tag-Free and Isotopomer-Selective Vibrational Spectroscopy of the Cryogenically Cooled H₉O₄⁺ Cation with Two-Color, IR–IR Double-Resonance Photoexcitation: Isolating the Spectral Signature of a Single OH Group in the Hydronium Ion Core”, *J. Phys. Chem. A* 2018, 122, 48, 9275–9284

From the Abstract of that paper “...the latter spectrum is dominated by a broad feature assigned to the isolated hydronium OH stretching fundamental with an envelope that is similar to that displayed by the H₃O⁺·(H₂O)₃ isotopologue. The feature appears with a diffuse band ~380 cm⁻¹ above it, which is assigned to a combination band involving the hydronium OH stretching vibration and the frustrated translation mode of the HD₂O⁺ core and one of the solvating water molecules. These trends are analyzed with anharmonic calculations involving four-mode coupling on a realistic potential surface A similar analysis was applied in Duong, C. H. et al. Gorlova, O.; Yang, N.; Kelleher, P. J.; Johnson, Disentangling the Complex Vibrational Spectrum of the Protonated Water Trimer, H₃O⁺·(H₂O)₃, with Two-Color IR-IR Photodissociation of the Bare Ion and Anharmonic VSCF/VCI Theory. *J. Phys. Chem. Lett.* 2017, 8, 3782–3789. This could also be cited as another example of the careful analysis done using VSCF/VCI Theory. Also the authors may wish to consult “Deconstructing Prominent Bands in the Terahertz Spectra of H₇O₃⁺ and H₉O₄⁺: Intermolecular Modes...” *J. Phys. Chem. Lett.* 2018, 9, 4, 798–803

3. Ref. 4 was not focused on an interpretation of the H₉O₄⁺ paper. The focus was on testing the TRPMD method for this cluster. Ref. 13 and the additional references noted above are far more relevant and important.

4. With regard to the role of Zundel in large complexes the authors probably cite and discuss “High-Level VSCF/VCI Calculations Decode the Vibrational Spectrum of the Aqueous Proton, Qi Yu,^{*,†} William B. Carpenter,[‡] Nicholas H. C. Lewis,[‡] Andrei Tokmakoff,[‡] and Joel M. Bowman^{*}, *J. Phys. Chem. B* 2019, 123, 7214–7224. There is a lot of analysis in this paper that really should inform the authors of the present paper. in particular the breadth of experimental bands of the true “aqueous proton”. From that paper we read “On one end, the excess proton localizes on one water to form a hydronium ion with a particularly tight first solvation shell, known as the Eigen species H₉O₄⁺.” Again the comment in the MS about a proton transfer mode in H₉O₄⁺ is at least misleading if not outright wrong. The authors need to address this.

5. The evolution of Zundel proton transfer mode with OO distance was analyzed in several earlier papers and these should be cited and discussed “How the Zundel (H₅O₂⁺) Potential Can Be Used to Predict the Proton Stretch and Bend Frequencies of Larger Protonated Water Clusters” *J. Phys. Chem. Lett.* 2016, 7, 5259 – 5265 and references therein.

In summary, while the calculations are indeed impressive their scientific impact needs to be more clearly fleshed out. This is not an easy task and I don't feel the authors have done an adequate job of doing this. Right now I'm not convinced that there is very much impact. The authors can certainly rebut this, but to do so they really need to read and digest the literature, as indicated above. In other words what new things have we learned from these calculations.

Reviewer #2 (Remarks to the Author):

This MS reports high level quantum MCTDH simulations of the vibrational spectra of the famous Eigen (H_9O_4^+) cation. The full dimensional (33D) simulation successfully reproduces the experimental vibrational spectrum and also confirms the spectral assignments of various bands as made in the VSCF/VCI spectra reported by Yu and Bowman (ref. 5, 13, 14). It is good that the authors did not focus on the spectral assignment which may look like a refinement over the established results. Instead, the authors conducted systematic analyses on the origins of the broad proton transfer band focusing on the specific roles of hydronium core and its solvation shell. Together with the simulation focusing on spectral contribution from only one hydronium + one solvation water (distorted Zundel subunit), the authors show that the Zundel subunit can provide sufficient spectral information to explain the anharmonic couplings and spectral features of the whole cation. Such treatment and conclusion can also be found in a previous paper by Yu et al. JPCB, 2019, 123, 7214. From the perspective of vibrational spectrum, these are important to help understand the role of hydrated proton where even the Eigen cation can be viewed as three overlapping and distorted Zundel subunits.

The calculations and analyses were performed systematically and carefully. Overall, this work is impressive. However, before I can recommend the publication in Nature Communication, there are several comments which the authors need to address carefully.

1) In the Abstract, line 27 of Page 2, the authors claim that "the large broadening of the proton transfer peak in the experimental IR spectrum of the Eigen cation, of which the origin remained so far unclear. [3-5]" However, the detailed spectral assignment of the broad proton transfer band at $\sim 2650 \text{ cm}^{-1}$ has been reported by Yu and Bowman in Ref. 5, 13 and 14. I did not see more detailed

spectral assignment by the current work. The authors also mention it on line 66-70 of Page 4. So the sentence in abstract can be overstatement.

2) On Page 3, line 56, the authors introduce the VSCF/VCI work, Ref.5, by Yu and Bowman, but did not discuss and distinguish the main difference between the previous VSCF/VCI and current MCTDH work. This makes readers think that the authors are working on the same problem just with a different method.

3) On Page 6, for figure 2.d, why are the intensities of two low-frequency peaks around 300 cm^{-1} so strong, almost comparable to the proton transfer band? Did the authors enlarge the intensities of these low-frequency bands?

A follow-up comment for Figure 2 is that the simplest 9D simulation (Figure 2.a) could result in the proton transfer band locating around 2600 cm^{-1} . It is known that the harmonic frequencies of proton transfer modes are around 3000 cm^{-1} . Does the inclusion of hydronium wagging and bending modes have significant impact to this 400 cm^{-1} red shift? Or will the proton transfer band locate at 2600 cm^{-1} when only three proton OH stretches are included?

3) On Page 7, line 105-111, I am confused by the set-up of corresponding MCTDH simulations. In line 105, the authors mentioned that certain coordinates of the Eigen cation were frozen to their equilibrium positions. However, in line 110, it is noted that specific coordinates were frozen to their expectation value in the ground vibrational state. The authors should give clearer technique details of the system set-up to avoid contradictory statements.

4) On Page 11, line 178-182, the authors noticed that there will be dramatically red shift of the proton transfer band when pulling the water molecule away from the hydronium core. Such behavior has been investigated in the literature for many times, but focusing on the OH distance and proton transfer band, (see Ref 51, Yu et al. JPCB, 2019, 123, 7214–7224, and Boyer et al. JPCL, 2019, 10, 918). Did the authors have a chance to check the expectation value of OH distance from ground state wavefunction and its correlation with reported red shift?

5) On Page S1 of Supporting Information, line 16, what does it mean by “about 2 cm^{-1} per mode”? My understanding is that the whole spectrum was shifted by 70 cm^{-1} not according to the specific mode.

6) There are some typos in the manuscript and the references.

For example, on Page 2, line 33-34, “charaterize” should be “characterize”, “this species” should be “these species”. On Page 11, line 172, “strickingly” should be “striking”

Reviewer #3 (Remarks to the Author):

This article provides a very accurate reconstruction of the absorption spectrum of the Eigen Cation using MCTDH. It is demonstrated that the spectra can be reproduced from the spectra of the Zundel as long as the the two other (frozen) water ligands are present. This is a very technically impressive calculation, and the agreement is very impressive. I don't have any technical concerns and think the paper is essentially publishable as is. I like the insights provided by the deconstruction of the spectra by freezing various modes.

I personally think this question of whether the proton forms an eigen or zundel cation has received too much attention relative to its practical importance. I have always thought that the only sensible answer is that it depends entirely on how precisely you define it and what the property of interest is, and that most commonly the answer is going to be some kind of mixture, so it is a relatively uninteresting question to me.

Indeed, in this case they have nicely demonstrated how both pictures are important to understanding the spectra and considering the impressive technical achievements of the calculation and the fact that the community seems to consider this an important question for some reason, I think it is worth publication in this journal.

Some minor comments:

Comparison should be provided with the latest Voth group paper on this: [10.1021/jacs.1c08552](https://doi.org/10.1021/jacs.1c08552)
Based on my understanding it seems like the two conclusions are consistent. They agree on the 'special pair dance' notion. But any disagreement should be clarified.

The authors state “This statement does not concern the relative population of the Zundel and Eigen structures in solution” But surely the authors results have implications for the solution phase? In particular, they imply that there is no way to distinguish them cleanly in solution as the eigen cation

is composed of Zundel subunits. But the other two water molecules are also integral to its behaviour?

Minor comments:

“...similar to the full Eigen cation spectrum, as seen in Fig. 4” The full Eigen spectrum isn’t in Fig 4 right? This is confusing phrasing.

Are all the plots in Fig 2 shifted? Or just f?

Authors' replies to the Referee reports on: The coupling of the hydrated proton to its first solvation shell

Markus Schröder, Fabien Gatti, David Lauvergnat,
Hans-Dieter Meyer, Oriol Vendrell

We are grateful to the three Referees for their time spent reviewing our manuscript, and for their critical and very insightful remarks. They have been tremendously helpful for us when improving the manuscript and addressing their queries and suggestions.

In summary, both Referees #2 and #3 qualify the work as “impressive” and recommend publication after we have addressed their queries. Referee #1 states that the paper is “state-of-the-art” but recognises some weaknesses that the Authors need to address. We agree with most remarks by Ref. #1, which are related to important context we missed from previous related works. We have incorporated these elements, some of them overlapping with comments from Referee #2, and we believe we have addressed them in full.

Below we reproduce the comments and queries of the Referees in full length using blue italics. Our replies follow in normal, black font.

Reviewer #1 (Remarks to the Author):

There is no question that these are state of art MCTDH calculations of the IR spectrum of H₉O₄⁺ using state-of-the art potential and dipole moment surfaces of Yu and Bowman and co-workers. However, the paper is weak in several key ways and these weaknesses need to be addressed.

First, the calculations are not put into proper scientific context in order to assess the scientific impact of the calculations. Indeed, there was controversy surrounding the interpretation (Zundel vs Eigen) of the experimental spectra of Johnson and co-workers of the H₉O₄⁺ cluster. The authors are apparently unaware that this controversy was settled in their ref. 13 “High-Level Quantum Calculations of the IR Spectra of the Eigen, Zundel, and Ring Isomers of H⁺(H₂O)₄” Find a Single Match to Experiment J. Am. Chem. Soc. 2017, 139, 10984–10987. Those calculations were carefully done using VSCF/VCI considering the Eigen, Zundel and Ring isomers and definitely found a match the Johnson experiment for the Eigen isomer. (See file attached) I guess this is conclusion of this paper. Well that's fine but it's not new. This is nice corroboration of that work. Figure 1 comparing the present calculations with experiment is impressive. However, a very similar comparison was already published using the same PES and DMS earlier in their ref. 13 and strangely this not noted. I reproduce a figure from the JACS paper below.

Also there was no shifting of that VSCF/VCI spectrum, whereas we read in Fig. 1 caption “Calculated spectrum (red-shifted 70 cm⁻¹ to match experimental line positions).

Author's reply: Reference 13 (new Ref. 6) is a very important landmark in the understanding of the Eigen cation through spectral simulations and analysis. Following their analyses, the authors of that work could establish, for the first time, that the previously measured IR spectrum (Science 2016) corresponded unambiguously to the hydronium form of the Eigen cation. We have discussed this context now when introducing this reference, and fully acknowledge its significance. We stress, though, that our manuscript was not, and is not, concerned with establishing this fact, which is merely corroborated by our full-dimensional calculations.

The authors need to discuss the cause of this 70 cm⁻¹ “offset” with experiment

Author's reply: The 70 cm⁻¹ shift stems from the fact that we obtain the ground state energy and the spectrum from separate calculations. The wavefunction of such a big system as Eigen is very difficult to converge. For

both calculations we try to find the best possible converged solution with our numerical resources. The ground state wavefunction has a much simpler structure than the time-evolved one and it is hence better converged. Therefore, the relative energies within the computed spectrum are more accurate than the absolute energies. Note that a shift is introduced for the full dimensional 33D calculation only, the reduced dimensional calculations do not require a shift. This is now explained in the paper where this shift is introduced. Note, the VCI calculations of Ref. 13 (new Ref. 6) are all of reduced dimensionality.

This important controversy should be highlighted in the paper; however, the fact that it has now been definitively resolved reduces the scientific impact of this manuscript. The authors go on to write “In particular, we identify the anharmonic vibrational modes that explain the large broadening of the proton transfer peak in the experimental IR spectrum of the Eigen cation, of which the origin remained so far unclear.3–5”. Well this is not accurate. See below

Author’s reply: As just mentioned above, this controversy, which is important within the literature of the hydrated proton and Eigen in particular, has now been contextualized in the discussion. It is, however, external to the main conclusions of this manuscript. Therefore, the fact that this debate is already settled, does not weaken our contribution. This manuscript’s main conclusion (backed by the 33-D simulations of the Eigen cation in curvilinear coordinates) is that the H_5O_2^+ subunit can explain both the spectral signatures of the Zundel (H_5O_2^+ in isolation) and Eigen (H_5O_2^+ polarized) cations in all their complexity. It is the smallest subsystem with this property, as the H_3O^+ subunit, either isolated or statically polarized, cannot account for the Zundel and Eigen spectra (both positions and line broadenings).

Also, there is no evidence of a proton transfer occurring in the Eigen cation H_9O_4^+ , quite the opposite. See below.

Author’s reply: We agree with the Referee and have changed all instances of “proton transfer” to “core O-H stretch”. The Referee is right that one should reserve the word “transfer” for situations where the proton is equally shared by both oxygen atoms and can be transferred from one to the other.

Some strange wording, 1. “The proton transfer band of the Eigen cation was studied by Yu and Bowman, 13 using a variational configuration interaction approach with up to a 4-mode representation, using normal modes, of the same PES we are using. . .”. Well that PES and DMS is the earlier Yu-Bowman PES and DMS that the authors are using. I suggest referring to the PES as the “Yu-Bowman PES” which was actually first reported in Yu, Q.; Bowman, J. M. Communication: VSCF/VCI Vibrational Spectroscopy of H_7O_3^+ and H_9O_4^+ using High-level, Many body Potential Energy Surface and Dipole Moment Surfaces. J. Chem.Phys. 2017, 146, 121102. This should be cited.

Author’s reply: The manuscript refers now to the PES as “Yu-Bowman PES” and also cites the suggested references.

2. Continuing “found that this band is characterized by highly entangled eigenstates in terms of normal-mode excitations, and that therefore these eigenstates cannot be assigned each to a well-defined vibrational configuration. This is not correct. That band was of course focused on in the papers using the Yu-Bowman PES and DMS in VSCF/VCI calculations. In particular, the authors need to cite and discuss “Tag-Free and Isotopomer-Selective Vibrational Spectroscopy of the Cryogenically Cooled H_9O_4^+ Cation with Two-Color, IR–IR Double-Resonance Photoexcitation: Isolating the Spectral Signature of a Single OH Group in the Hydronium Ion Core”, J. Phys. Chem. A 2018, 122, 48, 9275–9284

Author’s reply:

The Referee mentions here important aspects related to the assignment of the broad excess-proton band. We agree with the Referee and have undertaken the corresponding modifications.

We now introduce and discuss in more detail the old Ref. 13 (new Ref. 6), and here in particular Ref. 4 (Duong et al.), where the broad hydronium O-H stretch band was considered in detail. We remove the problematic statement mentioned by the Referee and we now write:

While studying the broad O-H stretch peak, Duong et al.4 found that this band is characterized by many highly entangled eigenstates in terms of normal-mode excitations. In the theoretical part of their work, Duong

et al. used VSCF/VCI calculations involving the hydronium core modes, O-O stretch and O-H bending modes to identify states contributing to the broadening.

From the Abstract of that paper “. . . the latter spectrum is dominated by a broad feature assigned to the isolated hydronium OH stretching fundamental with an envelope that is similar to that displayed by the H₃O⁺-(H₂O)₃ isotopologue. The feature appears with a diffuse band ~380 cm⁻¹ above it, which is assigned to a combination band involving the hydronium OH stretching vibration and the frustrated translation mode of the HD₂O⁺ core and one of the solvating water molecules. These trends are analyzed with anharmonic calculations involving four-mode coupling on a realistic potential surface. . . . A similar analysis was applied in Duong, C. H. et al. Gorlova, O.; Yang, N.; Kelleher, P. J.; Johnson, Disentangling the Complex Vibrational Spectrum of the Protonated Water Trimer, H⁺(H₂O)₃, with Two-Color IR-IR Photodissociation of the Bare Ion and Anharmonic VSCF/VCI Theory. J. Phys. Chem. Lett. 2017, 8, 3782–3789. This could also be cited as another example of the careful analysis done using VSCF/VCI Theory. Also the authors may wish to consult “Deconstructing Prominent Bands in the Terahertz Spectra of H₇O₃⁺ and H₉O₄⁺: Intermolecular Modes. . .” J. Phys. Chem. Lett. 2018, 9, 4, 798–803

Author’s reply:

We have noticed this diffuse band that the Referee is mentioning. As the Referee correctly points out, it has been the subject of previous analysis. This is now mentioned in the manuscript, and the corresponding references are cited. In the previous version of the manuscript we had indeed not paid much attention to this diffuse feature, but now we note that our simulations reproduce it very well, both in position and relative intensity:

Moreover, now the low intensity background on the high-energy shoulder at around 3000 cm⁻¹ emerges. VSCF/VCI analysis [4, 14] attributed this to a combination mode of hydronium O-H stretch and O-O stretching modes. This assignment is fortified in Fig. 2 c) where a peak at 3000 cm⁻¹ appears while only the hydronium core and the O-O stretches are modeled. Adding the ligand wagging modes then leads to the diffuse signal observed in the experimental spectrum.

3. Ref. 4 was not focused on an interpretation of the H₉O₄⁺ paper. The focus was on testing the TRPMD method for this cluster. Ref. 13 and the additional references noted above are far more relevant and important.

Author’s reply: We agree with the Referee and put now much more emphasis on introducing Ref. 13 (new Ref. 6) and the other suggested papers.

4. With regard to the role of Zundel in large complexes the authors probably cite and discuss “High-Level VSCF/VCI Calculations Decode the Vibrational Spectrum of the Aqueous Proton, Qi Yu,^{} William B. Carpenter,[‡] Nicholas H. C. Lewis,[‡] Andrei Tokmakoff,[‡] and Joel M. Bowman^{*}, J. Phys. Chem. B 2019, 123, 7214–7224. There is a lot of analysis in this paper that really should inform the authors of the present paper. in particular the breadth of experimental bands of the true “aqueous proton”. From that paper we read “On one end, the excess proton localizes on one water to form a hydronium ion with a particularly tight first solvation shell, known as the Eigen species H₉O₄⁺.” Again the comment in the MS about a proton transfer mode in H₉O₄⁺ is at least misleading if not outright wrong. The authors need to address this.*

Author’s reply: We thank the referee for pointing us to this set of references. We have added them to the citation list. We suspect that this comment has been triggered by our too broad use of the word “Zundel”. We agree that it is misleading to refer to the “Zundel” ion when discussing the polarized H₅O₂⁺ unit embedded in the Eigen ion, and we acknowledge that the role of the actual Zundel form (with a proton equally shared by two water molecules) has been studied in larger clusters. We are now citing all these works.

To avoid misleading the readers, we have changed all mentions of “Zundel” to “polarized H₅O₂⁺”, when this subunit is described in the presence of static, polarizing water molecules. We strictly reserve “Zundel” for the case when the excess proton is equally shared by two water molecules. In the polarized environment, the H₅O₂⁺ subunit does not feature a proton transfer mode anymore, as the Referee correctly points out, but a proton stretch mode with the same frequency as the Eigen ion.

5. The evolution of Zundel proton transfer mode with OO distance was analyzed in several earlier papers and these should be cited and discussed “How the Zundel ($H_5O_2^+$) Potential Can Be Used to Predict the Proton Stretch and Bend Frequencies of Larger Protonated Water Clusters” *J. Phys. Chem. Lett.* 2016, 7, 5259 – 5265 and references therein.

Author’s reply: We follow the suggestion of the Referee and discuss now these references. They are related to the point in our manuscript where we describe pulling out the polarizing water molecules from the dynamically active $H_5O_2^+$ subunit, which brings the spectrum in the direction of a Zundel (now really with an equally shared proton). The reference above analyzes the situation in which the two water molecules of the Zundel ion are pulled apart, leading to a modified potential for the shared proton that resembles the situation in larger clusters. Although the two scans are different (splitting the Zundel vs splitting the Eigen ions), they both exemplify how smaller subunits are embedded in larger ones. Hence the comparison is justified and has been added.

In summary, while the calculations are indeed impressive their scientific impact needs to be more clearly fleshed out. This is not an easy task and I don’t feel the authors have done an adequate job of doing this. Right now I’m not convinced that there is very much impact. The authors can certainly rebut this, but to do so they really need to read and digest the literature, as indicated above. In other words what new things have we learned from these calculations.

Author’s reply: We thank the Referee for praising our calculations as “impressive”. The main concern of the Referee was that, despite of this fact, the existence of previous works would diminish the impact of this manuscript. We have made our best effort to discuss more thoroughly and precisely the existing literature on the Eigen cation. This is not only fair, but enriches our work placing it in a broader context. However, we would like to reemphasize that the conclusions of this manuscript are independent of previous references and were not concerned with establishing the identity of the species measured in the seminal Eigen spectra recorded by Johnson (*Science* 2016), which was settled in a following up paper by Bowman (Ref. 6).

Indeed, Referee #2 below praises our work for not having focused on the assignment of the spectrum, which is largely well established in the literature, and instead having focused on a systematic analysis of the role of the various subunits and coordinates participating in the solvation shell.

Based on these arguments, and the modified and enhanced discussions in the context of the suggested references, we hope to have convinced Referee #1 of the independent and fundamental impact of our work.

Reviewer #2 (Remarks to the Author):

*This MS reports high level quantum MCTDH simulations of the vibrational spectra of the famous Eigen ($H_9O_4^+$) cation. The full dimensional (33D) simulation successfully reproduces the experimental vibrational spectrum and also confirms the spectral assignments of various bands as made in the VSCF/VCI spectra reported by Yu and Bowman (ref. 5, 13, 14). It is good that the authors did not focus on the spectral assignment which may look like a refinement over the established results. Instead, the authors conducted systematic analyses on the origins of the broad proton transfer band focusing on the specific roles of hydronium core and its solvation shell. Together with the simulation focusing on spectral contribution from only one hydronium + one solvation water (distorted Zundel subunit), the authors show that the Zundel subunit can provide sufficient spectral information to explain the anharmonic couplings and spectral features of the whole cation. Such treatment and conclusion can also be found in a previous paper by Yu et al. *JPCB*, 2019, 123, 7214. From the perspective of vibrational spectrum, these are important to help understand the role of hydrated proton where even the Eigen cation can be viewed as three overlapping and distorted Zundel subunits.*

The calculations and analyses were performed systematically and carefully. Overall, this work is impressive. However, before I can recommend the publication in Nature Communication, there are several comments which the authors need to address carefully.

1) In the Abstract, line 27 of Page 2, the authors claim that “the large broadening of the proton transfer peak in the experimental IR spectrum of the Eigen cation, of which the origin remained so far unclear. [3-5]” However, the detailed

spectral assignment of the broad proton transfer band at 2650 cm⁻¹ has been reported by Yu and Bowman in Ref. 5, 13 and 14. I did not see more detailed spectral assignment by the current work. The authors also mention it on line 66-70 of Page 4. So the sentence in abstract can be overstatement.

Author's reply: The referee is correct that we missed to properly cite and put into context the work of Yu and Bowman, in particular Ref. 13 (new Ref. 6). We also agree that this particular sentence in the abstract understates what was already known about the nature of the broad proton-stretch band in the Eigen cation. We have changed the abstract and the discussion within the manuscript. Besides identifying the role of the polarized H₅O₂⁺ subunit, one of the main findings of the present manuscript is that, besides the modes already identified by Yu and Bowman as well as in experiments, the *ligand* waggings play a very important role for the broadening of the core O-H stretching peak. We have pointed this out more clearly in the abstract and discussion.

2) On Page 3, line 56, the authors introduce the VSCF/VCI work, Ref.5, by Yu and Bowman, but did not discuss and distinguish the main difference between the previous VSCF/VCI and current MCTDH work. This makes readers think that the authors are working on the same problem just with a different method.

Author's reply: We have pointed out the differences to previous works by Yu and Bowman, in particular 1) full 33D calculations, 2) use of polyspherical coordinates adapted to the Eigen motif and 3) canonical polyadic decomposition of the PES to construct the sum-of-products potential operator, as compared to n-mode representations, i.e. our operator includes mode-mode couplings to all orders numerically.

3) On Page 6, for figure 2.d, why are the intensities of two low-frequency peaks around 300 cm⁻¹ so strong, almost comparable to the proton transfer band? Did the authors enlarge the intensities of these low-frequency bands?

Author's reply: All spectra in Fig. 2 are normalized to unit maximum height. In Fig. 2d) the core O-H stretching peak separates into multiple smaller peaks such that the relative height of the low frequency peaks increases. We have added a comment to clarify this in the manuscript.

A follow-up comment for Figure 2 is that the simplest 9D simulation (Figure 2.a) could result in the proton transfer band locating around 2600 cm⁻¹. It is known that the harmonic frequencies of proton transfer modes are around 3000 cm⁻¹. Does the inclusion of hydronium wagging and bending modes have significant impact to this 400 cm⁻¹ red shift? Or will the proton transfer band locate at 2600 cm⁻¹ when only three proton OH stretches are included?

Author's reply: We have conducted a calculation as in Fig. 2a) but only including the three hydronium O-H stretching coordinates while all other 30 coordinates are frozen. We observe three peaks between 2250 cm⁻¹ and 2600 cm⁻¹. The frequencies of the O-H stretching modes hence seem to be lowered by the presence of the oxygen atoms of ligand water molecules which weaken the core O-H bond. Interaction with the core wagging and bending modes seems to play a minor role for the red shift.

3) On Page 7, line 105-111, I am confused by the set-up of corresponding MCTDH simulations. In line 105, the authors mentioned that certain coordinates of the Eigen cation were frozen to their equilibrium positions. However, in line 110, it is noted that specific coordinates were frozen to their expectation value in the ground vibrational state. The authors should give clearer technique details of the system set-up to avoid contradictory statements.

Author's reply: The referee is correct: we have used inconsistent and not exact language. All coordinates that are frozen are set to their expectation value in the ground state. This has been corrected in the manuscript.

4) On Page 11, line 178-182, the authors noticed that there will be dramatically red shift of the proton transfer band when pulling the water molecule away from the hydronium core. Such behavior has been investigated in the literature for many times, but focusing on the OH distance and proton transfer band, (see Ref 51, Yu et al. JPCB, 2019, 123, 7214–7224, and Boyer et al. JPCL, 2019, 10, 918). Did the authors have a chance to check the expectation value of OH distance from ground state wavefunction and its correlation with reported red shift?

Author's reply: We have cited the respective literature and calculated the changes of expectation values of both, the O-H distance as well as the O-O distance. Indeed we observe a reduction of the O-O distance and

increase of the core O-H distance indicating that the H_5O_2^+ subunit becomes more Zundel-like. We have added these findings to the discussion in the manuscript.

5) *On Page S1 of Supporting Information, line 16, what does it mean by “about 2 cm-1 per mode”? My understanding is that the whole spectrum was shifted by 70 cm-1 not according to the specific mode.*

Author’s reply: The referee is correct, the spectrum is shifted as a whole. The statement “about 2 cm-1 per mode” is meant as a statistical average in the sense of “shift per involved coordinate/dimension”. This is clarified in the supporting information.

6) *There are some typos in the manuscript and the references. For example, on Page 2, line 33-34, “charaterize” should be “characterize”, “this species” should be “these species”. On Page 11, line 172, “strickingly” should be “striking”*

Author’s reply: The typos have been corrected.

Reviewer #3 (Remarks to the Author):

This article provides a very accurate reconstruction of the absorption spectrum of the Eigen Cation using MCTDH. It is demonstrated that the spectra can be reproduced from the spectra of the Zundel as long as the the two other (frozen) water ligands are present. This is a very technically impressive calculation, and the agreement is very impressive. I don’t have any technical concerns and think the paper is essentially publishable as is. I like the insights provided by the deconstruction of the spectra by freezing various modes.

Author’s reply: We thank the Referee for their very positive remarks on the technical and insight aspects of our work.

I personally think this question of whether the proton forms an eigen or zundel cation has received too much attention relative to its practical importance. I have always thought that the only sensible answer is that it depends entirely on how precisely you define it and what the property of interest is, and that most commonly the answer is going to be some kind of mixture, so it is a relatively uninteresting question to me.

Author’s reply: We agree that a sensible answer to these questions can only be given once the property of interest is well defined. Whether a precise answer is possible will depend crucially on the availability of theoretical descriptions of the clusters that are able to accurately reproduce experimental spectra. This, we are delivering in this work.

Indeed, in this case they have nicely demonstrated how both pictures are important to understanding the spectra and considering the impressive technical achievements of the calculation and the fact that the community seems to consider this an important question for some reason, I think it is worth publication in this journal.

Author’s reply: We thank the Referee for appreciating the “impressive technical achievements”, and for the positive recommendation towards publication.

Some minor comments:

Comparison should be provided with the latest Voth group paper on this: 10.1021/jacs.1c08552 Based on my understanding it seems like the two conclusions are consistent. They agree on the ‘special pair dance’ notion. But any disagreement should be clarified.

Author’s reply: We thank the referee for pointing us to this reference. Not only we have added the citation, but we have included the references to the works of two experimental groups that are discussed by Voth and coworkers. There is a recent debate about the relative populations of the Zundel and Eigen structures and this is mentioned in the conclusion. Our work does not bring information about these relative populations, but we have mentioned that our results are compatible with the idea of the ‘special pair dance’ model.

The authors state “This statement does not concern the relative population of the Zundel and Eigen structures in solution”

But surely the authors results have implications for the solution phase? In particular, they imply that there is no way to distinguish them cleanly in solution as the eigen cation is composed of Zundel subunits. But the other two water molecules are also integral to its behaviour?

Author's reply: As already said, we can not bring information about these relative populations, but we have added in the conclusion that "we stress that the difficulty to solve the problem may partly come from the fact that the H_5O_2^+ subunit can exhibit very similar spectral signatures compared to the Eigen cation when placed in a polarizing environment" as proven in our work.

Minor comments:

"... similar to the full Eigen cation spectrum, as seen in Fig. 4" The full Eigen spectrum isn't in Fig 4 right? This is confusing phrasing.

Are all the plots in Fig 2 shifted? Or just f?

Author's reply: Fig. 4 has been replaced by Fig. 1. In Fig. 2, only f has been shifted.

REVIEWERS' COMMENTS

Reviewer #2 (Remarks to the Author):

In the revised manuscript, the authors have addressed all my previous comments. With clearer presentation of background, methods/computational details and discussions, I believe this manuscript reveals how the vibrational spectral features of the hydrated proton can be affected by its environment and the associated mode couplings. The finding that H₅O₂⁺ subunit can describe both spectrum of the Zundel and Eigen cations is also insightful for future studies of hydrated proton systems. Therefore, I would like to support the publication of the current version of the manuscript in Nature Communication without further revisions.

Qi Yu